# After 55 Years of Neurorehabilitation, What Is the Plan?

**DOI:** 10.3390/brainsci12080982

**Published:** 2022-07-26

**Authors:** Hélène Viruega, Manuel Gaviria

**Affiliations:** 1Institut Equiphoria, Combo Besso-Rouges Parets, 48500 La Canourgue, France; helene.viruega@equiphoria.com; 2Alliance Equiphoria, 4, Résidence Le Sabot, 48500 La Canourgue, France

**Keywords:** neurological disorders, disability, activity limitations, participation restrictions, neurorehabilitation, neuroplasticity, homeostasis, functional recovery, quality of life

## Abstract

Neurological disorders often cause severe long-term disabilities with substantial activity limitations and participation restrictions such as community integration, family functioning, employment, social interaction and participation. Increasing understanding of brain functioning has opened new perspectives for more integrative interventions, boosting the intrinsic central nervous system neuroplastic capabilities in order to achieve efficient behavioral restitution. Neurorehabilitation must take into account the many aspects of the individual through a comprehensive analysis of actual and potential cognitive, behavioral, emotional and physical skills, while increasing awareness and understanding of the new self of the person being dealt with. The exclusive adoption by the rehabilitator of objective functional measures often overlooks the values and goals of the disabled person. Indeed, each individual has their own rhythm, unique life history and personality construct. In this challenging context, it is essential to deepen the assessment through subjective measures, which more adequately reflect the patient’s perspective in order to shape genuinely tailored instead of standardized neurorehabilitation approaches. In this overly complex panorama, where confounding and prognostic factors also strongly influence potential functional recovery, the healthcare community needs to rethink neurorehabilitation formats.

## 1. Introduction

Neurological disorders affect more than 1 billion people worldwide and are the leading cause of disability and the second leading cause of death [1,2,3]. These illnesses result from complex pathophysiological mechanisms, have diverse symptoms, variable time course and intricated consequences, including somatic, cognitive, emotional and social disabilities that limit daily activity and participation [3]. Most of these conditions, when affecting the individual moderately to severely, are very costly with socioeconomic implications due to direct health care fees, productivity losses, patients’ informal care issues (long-time sequelae management) as well as caregivers’ burden. Aside from therapy sessions, most of these moderately-to-highly disabled neurological patients spend most of their time isolated and inactive. Indeed, inclusive activities during daily living are generally unavailable even in the most developed societies. The mid-1960s was the official starting point for neurorehabilitation [4]. Since then, it has become key to question whether the rehabilitation environment, and not only the sophistication of the technical/technological approaches, is conducive to the patient’s recovery [2]. For instance, it has now been agreed that a lack of environmental enrichment negatively impacts functional recovery and might be a critical factor in time-dependent neurological decline [5]. The rehabilitative path is critical to both patients and their caregivers. Outpatient neurorehabilitation programs are quite dissimilar, ranging from specialized interdisciplinary programs (involving different configurations in the composition of the teams and techniques) to limited individual therapies [6]. In these conditions, quality issues may arise.

Patients must receive a current continuum program instead of patchy modules of care shaped by a specific purpose with separate accountability [7]. A fluid transition between each of these main elements is deeply needed. Moreover, a comprehensive understanding of the linkage between the nervous system, the body, the environment and the many issues shaping motor, sensory, cognitive and emotional functions is instrumental to provide appropriate rehabilitation solutions to neurologically disabled individuals. Interestingly enough, in a recent review of the top hundred cited neurorehabilitation papers from a database yielding more than 52,000 published studies between 2005 and 2015, only 14% focused on psychological symptoms, most frequently depression and anxiety, while another 5% covered cognitive functioning and only 4% focused on caregivers [8].

## 2. Searching Strategy and Selection Criteria

This review is based on our reading of existing medical literature in the web and a search of PubMed, MEDLINE, Cochrane Library, ScienceDirect and Google Scholar for articles as well as by handsearching reference lists. A few additional relevant papers were also found by hand in other grey literature sources. For searching, we used free language, mesh descriptors [mh], title words [ti] and authors [au]. Relevant search terms were combined with Boolean operators (OR/AND) as follows:

We determined a base criterion according to the following terms:

BASE_CRITERION (“neurorehabilitation”

OR “multidisciplinary rehabilitation”

OR “conventional neurorehabilitation”

OR “cognitive rehabilitation”

OR “neurological disorders”)

Then, to refine our search, we combined the base criterion with different sets of terms according to the chosen query as follows:

SET_ONE = BASE_CRITERION AND 

(“training intensity”

OR “dose and timing”

OR “dose response”

OR “repetitive task practice” 

OR “sensorimotor integration”)

SET_TWO = BASE_CRITERION AND

(“assessment and validation”)

SET_THREE = BASE_CRITERION AND

(“neurochemical changes”

OR “gut microbiome”

OR “circadian rhythms”

OR “motor learning”

OR “reward and neuroplasticity”

OR “neuroplasticity”

OR “fatigue”

OR “central fatigue”

OR “fuel metabolism”

OR “homeostasis”)

SET_FOUR = BASE_CRITERION AND

(“physical rehabilitation” 

OR “neuropharmacology”

OR “neuropharmacology side effects”

OR “neuropharmacology adverse effects”

OR “mirror therapy”

OR “constraint-induced movement therapy”

OR “functional electrical stimulation therapy”

OR “biofeedback”

OR “neurofeedback”

OR “brain-machine interfaces”

OR “robotics”

OR “neuroprosthetics”

OR “virtual reality”

OR “noninvasive brain stimulation”

OR “stem cells”

OR “cell therapy”)

SET_FIVE = BASE_CRITERION AND

(“active movement”

OR “intrinsic motivation”

OR “emotional intelligence”

OR “primary, secondary and tertiary prevention”)

We included original articles, trials, meta-analyses, reviews and a few case reports and editorials, all written in English, French and Spanish and published in peer-reviewed journals. More than 3000 articles were originally identified between January and June 2022, the abstracts were carefully read, and a shortlist of core published work of potential relevance to this review was prepared. Overall, 1030 articles were read by the two authors in order to provide a narrative review of the current neurorehabilitation landscape; also, reference lists of the retrieved articles were scrutinized for potentially relevant studies. In total, 885 articles were excluded from the review for different reasons including experimental work not confirmed in humans, neurorehabilitation propositions difficult to generalize, theoretical knowledge difficult to extrapolate, works with too small a sample size and works with very low quality evidence of improvement (see Figure 1).

## 3. Intensity and Effectiveness: A Simple Correlation?

Scientific evidence aiming to determine the minimum intensity and timing of neurorehabilitation is lacking and studies have biases of several kinds. So far, due to the very complex panorama of real world neurorehabilitation care and to irrelevant methodological procedures for clinical validation of therapeutical approaches in this field (i.e., standard procedures for drug candidates or medical devices; standard procedure for double-blinded and single-blinded trials), the effectiveness of each rehabilitation technique has not been clearly demonstrated for most of them. The studies carried out are mainly characterized by high heterogeneity due to confounding factors (mode of rehabilitation, degree of patients’ disability, quality of structures, types of evaluation) and low methodological quality [9,10]. For example, the increase of the standard deviation of a primary outcome variable due to recurrent intragroup variability in a stroke population (e.g., diverse pharmacological treatments for pain, seizures or mood disorders) inevitably lowers the power of a study. This example among others emphasizes the frailty of trial design in a real-world neurorehabilitation setting.

The substantial quantity of studies with small sample sizes reflects the difficulties that researchers in the field face in recruiting a large and homogeneous cohort from a given population displaying a myriad of neurological deficits’ configurations. Up to now, the “gold standard” determining the effectiveness of a health intervention for being clinically validated is a randomized controlled clinical trial protocol (RCTs). However, a growing number of researchers claim that this current standard procedure in not completely appropriate for studying complex interventions such as neurorehabilitation. Indeed, this comprehensive approach must be carried out according to specific ethical considerations (unlikelihood of suspending and/or delaying interventions or providing a “placebo procedure” in a control group), intragroup variability (heterogeneous populations with various clinical pictures and levels of disability), interdependent components and contexts, and comprehensive treatments involving organizational complexity and tailored multidisciplinary and cross-sectorial interventions. All these factors currently limit the methodological validity of the results [11,12,13].

The overall effectiveness of neurorehabilitation has initially been substantiated from old studies generally based on reduced mortality, lower dependency rates, lesser stay in a neurorehabilitation setting and/or decreased risk of institutionalization. In this relatively empirical and puzzling context, sessions of at least 30–60 min by type of rehabilitation, and global rehabilitation work up to half-day in moderate-to-severe disorders are most often recommended. The evidence related to the duration of therapy is questionable as it is limited by significant protocol heterogeneity that does not support robust conclusions [10]. 

It has also been claimed through the available evidence that more intensive early-onset neurorehabilitation (if the patient can bear it) promotes recovery [14]. The frequency is encouraged to be daily (or at least 5 days a week) and rehabilitation is to be carried out as long as improvement continues [10,14]. But is this actually realistic? Can an individual who has been confronted with the violence of substantial functional loss, whether at birth, abruptly or gradually in adulthood, retain the motivation and morale to work hard every day for barely perceptible results (the only people who have shown such everyday determination are high-level athletes)? If so, would it radically improve functional outcome? The lack of evidence in this area does not allow us to be conclusive.

Furthermore, the knowledge gathered in various complex areas of biology seems to diverge from the intensity/efficiency hypothesis. It is the case, among others, in the field of biological clocks, in that of learning, or in that of relationship between learning and fatigue or performance (physical or psychological) and fatigue. This is also the case for synaptic plasticity (at least partly) and its chronology, for metabolism and its long-lasting changes as well as for the foundations of homeostasis (changes depending on the available energy substrate). What do all these fields of knowledge tell us and how can we integrate them into a more comprehensive neurorehabilitation landscape in order to build a solid working base?

### 3.1. The Biological Clocks

Neuroplasticity is known to be subject to variability due to genetics [15], age [16], previous cortical activity [17] and attention [18], and those factors have shown to influence the magnitude of the plastic changes [19]. It has recently been suggested that circadian changes in various hormones may also play a key role [20]. Indeed, circulating levels of a myriad of neuromodulators (i.e., regulators or modifiers of electrochemical gradients and electrical impulses through the neural tissue) display circadian rhythmicity (i.e., their circulating level varies according to the time of day). These include cortisol, melatonin, thyroid stimulating hormone, dopamine, glutamate, GABA, prolactin, growth hormone and parathyroid hormone. Among these, cortisol is one of the most studied neuromodulators, the secretion of which occurs throughout the day, with a peak in the early morning in humans. Investigations have shown that both acute and chronic increases in cortisol levels, similar to those seen in psychological or physiological stress, impair learning and memory [19]. 

Circadian rhythms result in regulation of alertness, cognitive performance and fatigue throughout physical, mental and behavioral changes [21]. Motor function, neuromuscular efficiency, muscle fatigue, muscle recovery and general motor activity have been shown to follow similar circadian patterns [22,23]. In this context, it has been hypothesized that, at least in healthy individuals, postural control and gait are similarly influenced, leading to worst performance in the morning and peak performance in the late afternoon. These physiological behaviors are instrumental when designing neurorehabilitation protocols after a brain insult and should be considered to their full extent [24].

Circadian rhythms contribute to virtually all aspects of normal biological functioning. The behavioral and physiological programs of an organism are driven by a vast network of biological clocks (central and peripheral) and rhythms distributed throughout the body tissues. Consequently, each major physiological system (respiratory, cardiovascular, metabolic, endocrine, immune and nervous) follows circadian rhythms in its functioning [24,25]. The interactions within this dense network and its responses to external stimuli are highly complex. For example, they can vary considerably due to factors such as seasons, climate, social interactions or daily variations [26]. This implies that the organism adapts and anticipates through a flexible phenotype that responds to daily, seasonal and annual changes in its environment. This phenotype relies on a system of peripheral synchronization of the multiple clocks. A desynchronization of the different body clocks may be the result of stress and will cause significant disturbances of the circadian rhythms. This is probably a confounding factor that will promote or exacerbate metabolic and neurological disorders. Under these conditions, energy uptake and expenditure, as well as neuronal activation and inhibition, are unbalanced. Circadian dysfunction can be both a contributing factor and a consequence of the disease [25].

Since the aim of neurorehabilitation is to target neural remodeling based on the intrinsic flexibility of the system, it seems inconsistent to imagine a standardized neurorehabilitation intervention based on the intensity/efficiency paradigm excluding consideration of individual biological rhythms and clocks’ regulation of body functions, i.e., internal and external cues [24,25,26,27,28]. Endogenous circadian rhythms and their alignment with the external world offer an adaptive advantage by increasing energy efficiency that is difficult to manage once again through standardized procedures with a fixed temporality (e.g., 30–60 min sessions over half a day at the same time, at least 5 days a week). The effect of time of day on functional outcome would therefore depend on two main sources: (i) body’s internal clocks and other endogenous factors, and (ii) exogenous factors such as activity, mood and social interactions [23]. A comprehensive understanding of the influence of biological clocks in body’s functioning might help to optimize rehabilitation and training routines beyond the intensity/efficiency paradigm.

### 3.2. The Learning Brain

Learning of cognitive and motor tasks encompasses the (re)shape of neural representations linked to task implementation in everyday life and consequently is pivotal to neurorehabilitation [29,30,31]. However, learning is not a unitary process and occurs at different time scales, from minutes to even years. It relies on different brain systems and is controlled by specific mechanisms and behaviors. Whether or not in good health, there is no one-size-fits-all method to optimally train individuals. It is likely that each individual relies to varying degrees on their own unique learning mechanisms. The latter are influenced by many confounding factors, including age, motivation, experience and individual characteristics of the neurological disorder [32,33,34]. 

Memory needs a certain amount of time to be stabilized, which means that it is not generated immediately after the acquisition of new knowledge or skills. This stabilization involves two well-known processes: synaptic consolidation and systemic consolidation. The former is responsible for the transformation of short-term memory into long-term memory, and takes place in the first minutes to hours after the encoding of acquired information. The second is related to the post-encoding reorganization of long-term memory within brain networks and can take days, months or even years, depending on the memory network and the learned task [35,36,37]. Furthermore, consolidated learnings can be modified by memory reactivation. Thus, the processes of reactivation and reconsolidation are pivotal elements in the process of memory enhancement and neurorehabilitation re-shaping. This, in turn, has a potential impact in improving function-related performance [37].

For a long time, it was assumed that early high-dose longer-term training resulted in better patient outcomes [38]; however, recent studies have challenged this assumption [34,39]. Neurorehabilitation is not simply a matter of restoring a “normal function”. Newly learned patterns are more likely to persist when the resulting action is relevant to the patient. This means that therapy should target the deficit so that the resulting improvements are meaningful in daily life, i.e., outside the health institution. In this sense, little is known on how the nervous system prioritizes internally derived rewards (e.g., keeping postural stability, saving energy) over externally derived rewards (e.g., positive reinforcements provided by clinicians or therapists). Since very often the new learning skills depend on strong therapist support (or machine biofeedback) during rehabilitation, a considerable challenge consists of the sustainability of learned adaptations when the external feedback ceases once the patient returns to his daily life [34]. Consequently, it is crucial to know how to evolve towards an optimal application of the relearned actions to different real-life contexts (actions not envisaged during the rehabilitation).

The likelihood of updating and reinforcing learning (motor, cognitive,…) through reconsolidation of a skill after reactivation of the memory-related skill remains a very promising therapeutical option. Nevertheless, the knowledge required to confirm that reactivation and reconsolidation of different task-related functional memories could become a tool of choice in neurorehabilitation is still incipient since numerous fundamental questions remain unanswered. In the learning brain, specific actions are consistently associated with the reward mechanisms. Strengthening a desired action pattern is therefore not trivial since it requires an understanding of the type of reward that effectively reinforces the targeted pattern for a single patient. In this context, reinforcement-based mechanisms can be combined with other learning mechanisms to promote sustained improvement. Successful neurorehabilitation therefore must identify the key goal that is most rewarding to the patient and design approaches that reward the identified goal accordingly [34,37].

### 3.3. The Impact of Fatigue

The design of neurorehabilitation programs must also take into account other complex expressions of the disease that seem not fully compatible with intensive protocols. Among these, fatigue is a fairly disabling and frequent issue. Fatigue can be defined as a difficulty in starting or sustaining voluntary activities. 

Chronic fatigue is a typical symptom of neurological diseases with a rather high prevalence depending on the pathology (e.g., up to 77% after stroke, 75% after traumatic brain injury, 97% after brain tumors and 80% after multiple sclerosis) [40]. It can be classified as physical or mental, and the former can be categorized as peripheral or central. Peripheral fatigue is commonly identified with muscle fatigability due to disorders of the muscle cells and/or the neuromuscular junction, whether permanent or not. Indeed, physical inactivity and denervation modify the muscular phenotype, the muscle fibers becoming predominantly fatigable [41]. Central fatigue may be present in disorders of the peripheral, autonomic and central nervous system, and is commonly observed in lesions of the pathways associated with arousal and attention, the reticular and limbic systems, and the basal ganglia. However, it is not simply a feeling of physical exhaustion. Central fatigue also has a significant cognitive component (mental fatigue). Increased perception of effort and limited stamina of physical and mental activities are the main characteristics of central fatigue [40,42,43,44].

During motor or cognitive activity, the work output varies according to the internal environment (homoeostasis and autonomic function) and external environment (light, temperature, etc.), which can act as facilitators or inhibitors. Intrinsic and extrinsic motivation input acts at two levels during action: helping initiation of the voluntary effort (or its withdrawal) and modifying the internal environment control. Finally, work output is also tuned by feedback from motor, sensory and cognitive systems that determines the level of perceived effort [40]. An alteration at one of these levels will have a direct impact on the objective and subjective evaluation of the state of fatigue that the individual can achieve, having a potentially significant negative impact on the neurorehabilitation process [45].

The source of the fatigue is multifactorial and may arise from the neurological disease itself, an integral feature of the disease, the result of psychic disturbances during the course of the disease, or a side effect of the pharmacological treatment of the disease or its symptoms. Both peripheral and central mechanisms are implicated [46]. Management must be tailored, taking into account physical, cognitive, emotional and environmental cues. According to several studies, graded physical activity programs are recommended and have shown stable and overall better functional status characterized by fewer symptoms of mood disorders, and improvement in both sleep and physical endurance. The amount of activity should be limited and spread throughout the day, with the emphasis on regularity, not performance level. Cognitive compensation strategies that circumvent the limited energy resources available to neurological patients with fatigue may also be beneficial. These compensatory strategies require better planning and variation of activities to promote a more regular pattern of activity and rest. Emotional management must consider motivation, pleasure, self-confidence and social activity. Finally, environmental cues should focus on some key characteristics such as amenity, accessibility, safety and attractiveness [40,44].

In neurological diseases, the contribution of physiological, cognitive and affective changes underlying fatigue is variable and is largely at odds with the intensity/efficacy hypothesis.

### 3.4. The Impact of Psychological Factors

Psychiatric symptoms, mostly anxiety, depression and negative thoughts, are concomitant pathologies present in several neurological disorders with variable prevalence. Patients presenting those concomitant pathologies would have an increased risk of mortality, a higher risk of suicidality, increased cognitive impairment, increased risk of falls, increased hospitalization costs, a poorer quality of life, and decreased daily functioning and social interactions [47,48]. Additionally, the results of several studies suggest that those symptoms can directly have a negative impact on the neurological deficits leading to significant deterioration of the functional status of the patient [47,48,49,50,51,52,53,54]. However, psychiatric disorders may go unnoticed and have a significant influence on motivation and hence on the rehabilitation process.

For instance, a frequency of post-stroke depression around 30–35% and anxiety around 18–25% has been reported in previous studies [47,55]. The prevalence of anxiety disorders in multiple sclerosis is estimated to be around 30%, three times higher than in the general population. In these patients, the most common forms are generalized anxiety disorder (~18%), panic disorder (~10%), and obsessive-compulsive disorder (~9%) [54]. Anxiety disorders occur in approximately 25–35% of Parkinson’s patients. Panic disorder, generalized anxiety disorder and social phobia are the most common anxiety disorders reported in Parkinson’s disease and have been argued to be one of the earliest manifestations of the disease [50,51,53]. Moreover, there is a frequent co-morbidity between anxiety and depressive disorders in those patients, ranging from 14 to 26% [49]. Finally, reported rates of depression following traumatic brain injury vary widely from 17% to 61%, depending on, among other factors, cause and severity of the brain trauma, and methods used to assess it [52].

Impaired modulation of neural networks may be responsible for modification of functional outputs through an overload of the capacities of some key brain hubs to process competing inputs coming simultaneously from sensorimotor, cognitive and emotional activity [53]. Indeed, functional connections between limbic and motor circuits allow emotional inputs to interfere with motor outputs [51]. These phenomena could be responsible for gait freezing in Parkinson’s disease [50,51,53] or postural instability due to improper neuromuscular control in stroke [48]. For instance, anxiety may impair visual information processing as well as the interactions between visual, vestibular and somatosensory inputs, which are key components of postural control. In addition, high levels of anxiety lead to an attentional bias towards non-task related stimuli, resulting in a decrease in attentional resources allocated to the task and a decrease in performance [48].

A key issue in this context is that the impact of psychological factors must be taken at all times into account (and reevaluated periodically) when assessing functional outcomes and implementing different interventions to improve the functional abilities of neurological patients.

### 3.5. Reconfiguration Costs and Whole-Brain Homeostasis

The brain is a costly system to build and to run, topologically organized throughout neural networks in order to produce high value for low cost. Brain networks are shaped to have high topological efficiency, robustness and modularity [56]. The brain shows high global efficiency of information transfer between brain regions located far away from each other in the anatomical space [57]. The metabolic costs of the brain are disproportionate in relation to its size, i.e., 10 times higher than what would be expected from its weight alone compared to the body’s weight. However, they are thoroughly controlled to be as low as possible for any given function [58,59]. Most of the substantial brain’s metabolic costs are engaged in the active maintenance of electrochemical gradients across neuronal membranes [58]; these metabolic costs are pulled down by myelination and pulled up by axonal length and diameter, long-distance connections being metabolically more costly to maintain [60]. Furthermore, by minimizing the length of anatomical connections in the network (i.e., pulling down the wiring costs), the system will also regulate running-dependent costs. As mentioned before, neural networks typically have high overall efficiency [57]. This means that a trade-off between efficiency and connection distance can be quickly renegotiated by the brain depending on the complexity of the task. For example, when there is a great demand for cognitive processing, neural networks adopt a more efficient but more costly workspace configuration. Conversely, when cognitive demand is lower, the brain networks move to a less costly configuration [61].

As one would expect from the above, the brain is highly vulnerable to any condition that threatens its energy supply. In a pathological situation, main degenerative mechanisms converge to loss of intracellular homeostasis with subsequent energy failure and cell death. Neuronal survival is multifaceted and encompasses, among others: well-fueled energy metabolism, trophic input, clearance of toxic substances, appropriate redox environment, integrity of blood–brain barrier, regulation of programmed cell death pathways, and cell cycle arrest. Mechanisms of delayed degeneration entangle excitotoxicity subsequent to hyperexcitation of glutamatergic receptors, loss of intracellular calcium homeostasis, energy failure, endoplasmic reticulum stress, reactive oxygen species production, neuroinflammation and axonal degeneration of both synaptic inputs (anterograde degeneration), and projection targets (retrograde degeneration) [62]. On the other hand, intrinsic mechanisms of repair are also triggered by the nervous system in pathological conditions [63,64]. These molecular mechanisms of energy supply although mandatory for supporting brain plasticity, are mostly insufficient to preserve the overall tissue viability and to allow post-lesion functional restitution.

Several questions arise from the fundamental intuition of Ramon y Cajal in 1899 [65]: “All of the various conformations of the neuron and its various components are simply morphological adaptations governed by laws of conservation for time, space and material”. These “conservation laws” are probably at the origin of the final functional outcome relying on neuroplasticity. However, if the costs of reconfiguration compromise the whole-organism homeostasis, neurological restitution through neural plasticity could be seriously jeopardized. There are some key unknowns in the neural building and running landscape. Indeed, very few data exist on the energetic cost of potential de novo neural tissue development. We only know that, during development, it corresponds to 75% of the global caloric intake, which is colossal [66]. We also do not know with certainty the needed energy for the circumscription of an injury and the washing of cellular/molecular waste, nor the energy cost necessary for neuronal re-encoding and reshaping of redundancy and degeneracy functions. Finally, we know very little about the energy requirements for the establishment of mid- and long-term synaptic plasticity and appropriate functional restitution. In this context, the intensity/efficiency paradigm remains hazardous since it would not be compatible with the rules of homeostasis as managed by the brain. Moreover, it would also be conflicting with a reorganization of neural networks despite the extraordinary redundancy of the brain architecture (neuroplasticity), because of unavailable energy substrate.

## 4. The Challenge of a New Paradigm

In practice, the intensity with which rehabilitation is carried out varies among dispensing structures relying on historical choices, philosophy and preferred adopted individual/institutional methods. In the meantime, the magnitude of burden of neurological disorders compels governments and public policy makers to take urgent action to mitigate their risks and impact. However, a number of factors other than clinical efficacy may have an impact on the adoption of innovations and the optimization of care: for example, access to care, cost, medical education and degree of cooperation between different healthcare providers [3,67,68]. A clear panorama of this issue is nevertheless beyond the scope of this review. The functional recovery sought through neurorehabilitation is possible through a partially acknowledged brain reorganization thanks to the phenomenon of plasticity which would allow, in the best hypothetical case, a recovery without compensation [69].

Clinical researchers generally agree that it is problematic to select meaningful and appropriate outcome measures that reflect functional gains and recovery in patients with neurological disorders. Neurorehabilitation is a multifaceted intervention in which the active constituent is not always accurately identifiable. In addition, the floor/ceiling effects of clinical scales may limit sensitivity in measuring both patient’s status and treatment outcomes. While patients exhibit patent clinical heterogeneity and confounding factors, clinicians may have real difficulty incorporating the patient/caregiver perspective into decision making, and many issues essential to patients may be overlooked [2]. In this context, randomization cannot realistically and systematically eliminate potential confounders, even though they can be taken into account in the final analysis.

The key issue in this area of research is therefore to identify all or as many potential confounders as possible at the outset of a clinical trial. A more refined set of baseline measures is needed to characterize the full range of impairments and disabilities. This will allow, on the one hand, the implementation of improved randomization techniques beyond standard stratified procedures, and, on the other hand, a comprehensive evaluation of the effectiveness of a rehabilitation approach focusing on impairments, activity limitations and participation restriction, as advocated by the WHO [9,70].

The roadmap aiming to obtain maximum effectiveness of a neurorehabilitation treatment of patients with moderate-to-severe neurological disabilities should take into account, as a minimum, an earliest possible start of the treatment, sufficient duration and intensity, multidisciplinary and cross-sectoral integrative processes at different stages, systematic assessment/monitoring of the patient’s somatic, cognitive and psychological condition, and systematic problem-solving support for ADLs [71,72]. Additionally, a careful definition of adjustment variables and their timing should be incorporated. All these conditions, although necessary, are probably not sufficient for an optimal outcome in the standardized context provided by RCTs [73].

## 5. The Promised Revolution of New Technologies: A Real Opportunity or a Headlong Rush?

Formally speaking, neurorehabilitation encompasses a series of methods that assist individuals who experience disability (impairment, activity limitations, and participation restriction according to WHO) to achieve and maintain optimal function in interaction with their ecosystem, for maximum autonomy and social reintegration (relying on the biopsychosocial model) [2,74,75].

The continuum in the development of science has led in recent years to the integration of new technologies in the field of disability and rehabilitation. However, the validation of these cutting-edge approaches has followed the same path as their precursors. Thus, given the limited number of robust and methodologically sound studies, the quality of evidence is considered low for a majority of approaches including, among others, tele-rehabilitation, electro-mechanical and robot-assisted training, virtual reality, serious gaming (interactive video games) or whole-body vibration therapy [2].

The issue here is twofold: on the one hand, the benchmarking for an effective therapeutic result of such a multifaceted intervention has not been fully resolved in its conventional shape, and on the other hand, the new technologies and their effectiveness in the neurorehabilitation domain are being evaluated using the standard but undoubtedly inappropriate methodology.

The landscape of the neurorehabilitation modalities is broad and each approach has its own specific objectives, targets and drawbacks. The cornerstone of the whole neurorehabilitation system remains the conventional approach (physical therapy, occupational therapy, cognitive therapy, psychological support). It aims at promoting overall recovery of somatic (e.g., gross and fine motor control as well as interventions that have a more targeted aim, e.g., postural control, gait improvement, control of voluntary movements functions of the upper and lower limbs, abnormal movements control, regularization of muscle tone, among others) and cognitive functions (e.g., attention, concentration, memory, comprehension, communication, reasoning, problem solving, planning, initiation, judgement, self-monitoring and awareness). However, the chosen neurorehabilitation protocols vary radically according to fairly unobjective choices leading to the adoption of some methods rather than others. In general, the patient is supposed to fit in a standard procedure not necessarily adapted to her/his own needs, objectives and internal physiological rhythms. In this context, evidence to either support or refute the hypothesis that conventional rehabilitation interventions improve the targeted function is of insufficient quality [10,76,77,78].

Furthermore, adjuvant drug therapy is used for associated symptoms such as spasticity, seizures, sialorrhea, urinary retention, sleep disorders, neuropathic pain, anxiety, depression and former behavioral disorders. The used compounds have, independently, potential central nervous system side effects that can limit to varying degrees the functional capacities of the individual. The most common are fatigue, lethargy, somnolence, dizziness, unsteadiness, visual impairment, headache, postural hypotension, weight gain, psychomotor impairment, cognitive impairment, confusion, mood disorders, emotional lability, and even personality changes [79,80,81,82,83,84,85]. Moreover, combining two or more drugs places the patient in a grey zone where it becomes quite difficult to determine her/his real neurological potential and the actual disease-related symptoms. Lastly, some of these compounds potentially affect neuronal and synaptic plasticity, which is in all likelihood the foundation of neurorehabilitation [30,86].

Relying on conventional rehabilitation, some methods such as mirror therapy and constraint-induced movement therapy have expanded the catalog of modalities. In contrast to other interventions, which employ somatosensory input to assist motor recovery, mirror therapy is based on visual stimulation. During mirror therapy, a mirror is placed in the person’s midsagittal plane, thus reflecting the non-affected side as if it were the affected side. By this setup, movements of the non-affected side create the illusion of normal movements of the affected side, providing some kind of access to a body part that is otherwise not accessible. However, despite a trend toward a positive outcome, the working mechanisms are still unclear and most of the evidence is from studies with weak methodological quality, i.e., no clear conclusion could be drawn if mirror therapy replaced other interventions for improving motor function of the arm or leg, or both [87,88].

On the other hand, constraint-induced movement therapy consists of using exclusively the affected side by restraining the unaffected one during dedicated targeted exercises or usual activities of daily living. However, this therapeutic technique is difficult to deliver because it requires proper allocation of resources by both clinicians and patients (around 70% of patients seem not to be interested in participating because of time constraints). In the latest systematic review and meta-analysis of constraint-induced movement therapy studies during a period from 1966 up to 2010, a trend to a limited improvements in motor function was disclosed, but these benefits do not convincingly reduce demonstrated long-term improvement in patient disability [89,90].

Technological advances have made it possible to imagine the replacement or reinforcement of physiological signals (depolarization of motor axons, synaptic current-dependent neurotransmitter release) to externally drive a function that cannot be appropriately controlled by the central nervous system. Developed in the sixties, functional electrical stimulation uses electrical current to produce contractions of paralyzed or paretic muscles in a sequence that allows performing tasks such as bringing a glass of water to the mouth, holding a toothbrush, grasping a key, standing, or walking. It potentially reduces disability by enhancing recovery of volitional movement (therapeutic effect) or by assisting and/or replacing lost volitional movement (neuroprosthetic effect). This technology is applicable to patients whose paralysis or paresis is caused by upper motor neuron injury and requires lower motor neurons that target muscles to be intact. Stimulation of the neuromotor system at different levels presumably impacts neuronal circuits and results in neuroplastic changes by rewiring the lesioned brain and allowing functional recovery. However, while a promising technology, most efficacy studies conducted to date on functional electrical stimulation are generally small in size and consequently limited in power to draw strong conclusions [91,92,93]. Moreover, since the therapeutic effects of functional electrical stimulation depend on training time and intensity, a critical limitation of available systems for motor neuroprostheses in the context of a rehabilitation program is the rapid onset of muscle fatigue [94,95]. Additionally, selective activation of muscles is another major challenge for clinical application [96].

In the same vein, non-invasive brain stimulation applies electrical or magnetically induced currents through the scalp to temporarily excite or inhibit brain activity in target regions. It may enhance functional recovery through the induction of neuroplastic mechanisms. Long cortical stimulation periods can produce lasting effects on brain functions, opening the way to their therapeutic use in the field of chronic neurological diseases. Emerging research suggests that different brain stimulation techniques including the most commonly used (transcranial magnetic (TMS) and direct current stimulation (tDCS)), would be technical adjuvants to conventional approaches secondarily impacting neurorehabilitation processes [97]. Brain stimulation causes both local changes in activity at the stimulated site and remote changes through different neural networks [98]. The latter is instrumental in knowing that a successful behavior requires the concerted action of multiple brain regions. Indeed, even though individual regions perform specific functions, the sharing of information amongst a wide array of interconnected regions is crucial for everyday complex behaviors. In this sense, the ability of non-invasive brain stimulation to modulate activity locally and in interconnected networks seems valuable for therapeutics [97,99]. However, there is no simple relationship between the excitability of a region and the activity in that region [98]. Thus, interpretation of state dependent effects is always speculative since the response changes according to the state of the cortex when the stimulus is applied. The use of non-invasive brain stimulation with other forms of brain-computer interfaces, robotic prosthetics, or even with enhancement of pharmacological treatments could yield superior gains. However, some gaps have been identified that still prevent some of these techniques from moving faster to clinical practice. These include complex and laborious application, suboptimal accuracy and reliability, and insufficient understanding of the neurobiological substrate of the responses. All these aspects need to be reasonably addressed for the methods to become clinically applicable and enter the diagnostic and therapeutic arena [100]. Moreover, recent analyses disagree on the effectiveness of these techniques in some neurological disorders and reported major adverse effects [101,102,103,104,105]. Finally, there is no comprehensive agreement on the optimal combination of non-invasive brain stimulation parameters, the duration of the treatment, and clinical outcome measures to evaluate the efficacy of the therapeutic intervention.

Physiological signals have also been used to offer a measurable and meaningful feedback to the patient in order to allow her/him to learn a more effective way to control her/his disabled body segments. Accordingly, augmented information on biomechanical or physiological parameters obtained by measuring body movement and force (biofeedback), or neurological parameters (biofeedback, neurofeedback) through non-wearable or wearable sensors generate a feedback signal clearly readable by the patient [106]. A biofeedback signal is based on mode (visual, auditory, vibrotactile, or a mix), content (information provided to the user grounded or not on performance or results, i.e., execution or consequence of the movement respectively), frequency (number of signal event occurrences per unit of time), and timing (at what time the signal is given with respect to movement execution). Biofeedback can be used for positive reinforcement (increase a specific movement pattern) or negative reinforcement (reduce a specific movement pattern) [107,108,109]. On the other hand, neurofeedback, sometimes referred to as EEG biofeedback, targets the brain and cognitive functions through the use of electroencephalography. It aims to restore or adapt the affected function (attention, concentration, memory, comprehension, communication, reasoning, problem solving, planning, initiation, judgement, self-monitoring and awareness) through the use of EEG and brainwave activity, providing a visual or auditory cue to the patient who can consciously adapt her/his brainwave activity to reach the targeted training threshold [110]. Most studies conducted so far in neurorehabilitation through biofeedback carry an undefined risk of bias. Moreover, the high variability in terms of type and configuration of sensors, outcome measures and modes makes it difficult to compare them in terms of effectiveness in specific populations. For instance, in stroke, a small amount of evidence from individual studies supports the use of biofeedback combined with standard physiotherapy. Indeed, improvements in motor power, joint range of motion, functional recovery and gait quality are not clearly beyond those of standard physiotherapy alone [107]. Additionally, in neurofeedback, investigations provide a low level of evidence and satisfactory methodological design. In stroke, modest positive improvement of a number of cognitive domains following neurofeedback has been noticed. Despite more than two decades of neurofeedback research, methodological quality and level of evidence has not substantially improved [110]. Due to the lack of evidence, there is still a need for optimized comparative studies to determine the best type of feedback according to the neurological disorder.

Beyond the physiological framework, a new level has been reached when neurorehabilitation became aware of systems that could partly or fully replace (wearable devices) or strengthen an impaired function (wearable devices and platform-based devices) by means of robotics. The use of robots theoretically improves neural plasticity by providing the patients with adaptive, repetitive, intensive and quantifiable task-oriented physical treatment. In this context, robotic aids must adapt to the status and recovery level of the patient [111,112]. The last decade has seen a significant increase in the use of robotic aids in neurorehabilitation. Compared to conventional care, robotic-based neurorehabilitation can in theory be performed more easily at high intensity and frequency, concurrently monitoring the achieved patient’s performance to better adapt the treatment level. It also can hypothetically generate fine-tuned actions during the task-oriented training, i.e., optimal movements and forces. Robotic-assisted models are meant to be developed to predict the use of different motor control strategies and to tailor treatment to the patient’s possibilities [12,113,114,115,116]. However, the perspective of the end user is crucial, and the robotic rehabilitation effectiveness will almost certainly depend on whether or not the patient sees it as a suitable path or an added value. Additionally, in spite of the outstanding technical developments in the field, evidence on how robotic technologies can positively impact patient’s recovery is still lacking. It is particularly unclear which intervention is most effective depending on the pathology and on the individual needs. Studies to date are based on low levels of evidence and conclusions should be considered with caution [111,112,114,115,116,117].

Computer-assisted programs and, more precisely, virtual reality have been integrated, like robotics, into the broad landscape of neurorehabilitation. Virtual reality artificially reconstructs life-like environments providing a persuasive perception of being inside a real situation that nonetheless relies on a computational model. Nowadays, this field continues to grow thanks to progress in high-quality low-cost motion capture and powerful cloud computing, allowing the technology to be more accessible to clinicians [118,119]. The technology is almost exclusively applied as an adjunct to conventional approaches of cognitive or neuromotor rehabilitation and is typically characterized by the use of gamified exergames which feature game-oriented exercises, although it can also be, for example, task-oriented in an art-therapy environment [120] or resembling the demands of everyday life activities [121]. Theoretically, the supposed advantage of virtual reality is the way in which motivational strategies in the form of gamification can be incorporated into training paradigms [122]. This would enhance patient adherence to high-dose repetitive functional training providing a welcome break from the monotony of therapy and increasing exercise performance [118]. Current evidence suggests that it could be useful in stroke [123], Parkinson’s disease [124], or multiple sclerosis [125] and would have a potential to be integrated in a home setting personalized rehabilitation program. However, specific neural mechanisms underlying cognitive and somatic improvement remain, to date, unknown [119]. Evidence for the efficacy of this technology is sparse, with some reviews suggesting an equivalent improvement to that of conventional approaches [118,123]. There is consensus that the quality of the data found in neurorehabilitation trials is limited (small sample sizes, high risk of bias because of poor reporting) [123,124,125,126]. A greater body of supporting evidence to assess the efficacy of this neurorehabilitation modality is still needed [127].

Finally, personalized regenerative medicine is gradually taking its place in the vast panorama of neurorehabilitation tools. Biological therapies for regenerative medicine focus on protective and regenerative strategies using stem cells and neural progenitor cells. They target the sub-acute and chronic phases of a neurological disorder and raise the hope of being the key for true neuro-restoration. Stem cells are defined as cells that can both self-renew and differentiate into other cell types for cell-based regeneration of damaged brain tissues and repair of neuronal structures leading to recovery of function [128]. It has been agreed that transplanted cells can provide benefits through various mechanisms, including local (cell replacement, trophic support, immunosuppression/anti-inflammation) and at distance phenomena (stimulation of endogenous signaling for neuroplasticity/regeneration, regulatory interactions with key support cells of the neuroglia such as astrocytes, microglia and oligodendrocytes) [129]. Stem cell transplantation has been applied in clinical trials for a variety of neurological diseases because of the regenerative and unlimited proliferative capacity of progenitor cells. Depending on the neurological pathology (e.g., cerebral palsy, Parkinson’s disease, stroke), different types of stem cells can be used and will have specific therapeutic effects [128,129,130,131]. Various cell sources show promise, although each has several problems that must first be overcome. There are several issues that need to be addressed before establishing stem cell therapy as a routine. These concern, among others, the type and number of stem cells to be grafted, the time of transplantation, the need of dose-boosters, or the route of administration [128]. However, since the doses, sources and routes differ among clinical trials, it is not clear to date which is the best candidate according to the pathology and/or the deficit. Additionally, the follow-up observation period of current clinical trials is usually short, which impedes a consistent assessment of long-term safety and efficacy of stem cell therapy. Great differences in population characteristics and evaluation methods among the published clinical trials is also noticed [131]. A second major issue in this field is the possibility of adverse events and serious adverse events. Indeed, the patient’s own inflammatory processes can lead to graft survival failure, bleeding, altered immune response, and can reduce or even abolish the therapeutic effects of transplantation. Accordingly, a potential obstacle to stem cell transplantation is the rejection of the transplanted cells (allogeneic stem cells) by the host immune system. Rejection may require the use of an immunosuppression protocol for continued benefit, which can have a detrimental effect on endogenous healing. In addition, immunosuppression may increase the risk of tumor formation. Finally, one of the most important concerns in cell therapy is the predisposition of transplanted cells to form tumors per se. Last but not the least, more complex conditions such as co-morbidities can affect the efficacy and effectiveness of a cell therapy and become extra clinical challenges [129]. Overall, long-term assessment of representative cohorts is mandatory.

Interestingly, one of the arguments frequently used for supporting some of the reviewed technologies (e.g., robotics or virtual reality) is the fact that the devices allow the health care provider to augment the dose of task-oriented exercises delivered to the patient. However, a proof of dose-efficacy in neurorehabilitation is still missing and the methodological approaches of the neurorehabilitation research are still biased.

## 6. Discussion

Experimental and theoretical neurobiological models and clinical practices have dramatically widened our knowledge of the brain functions and its vast and sophisticated intrinsic capabilities for managing disruption and adapt to external requests. They have revealed the complexity of the entanglement of the somatic, cognitive, motivational and emotional spheres. However, so far, no model has been able to comprehensively capture the mutual influence exerted by different processes across the conventional boundaries that segregate action, cognition and emotion [132]. 

Fifty-five years of neurorehabilitation have provided us with amazing tools for helping disabled individuals in their everyday lives. However, despite the deep knowledge of the nervous system and the sophisticated devices and methods that we have at our disposal, a ceiling effect is still noticed with respect to functional recovery. A consensus is gradually rising among the healthcare and scientific communities concerning the need to reformulate the problem. It is clear that the concerned healthcare community has to take into account the main topics that are considered essential by the duo patient/therapist for the enhancement of patient’s functional skills and quality of life in a practical neurorehabilitation setting. For instance, some studies have shown that motivation and attention share the same neural networks and that providing motivating experiences could account for an increased attention of the individual towards a specific rehabilitation program theoretically promoting treatment adherence and compliance [119].

Bearing in mind the different aspects likely to interfere in the rehabilitation process and the complexity of the many overlapping aspects of human being, the neurorehabilitation of the future is probably far more human than expected, even if it relies on the stronger points of neurorehabilitation methods and innovative developments. The rehabilitation approach needs to remain rigorously multidisciplinary and intersectoral, which requires a huge structural effort of health organization (hospital services, clinics, and city medicine) and an assumed systemic orientation. The patient’s long-term regular follow-up should be structured around the three pillars of health, primary, secondary and tertiary prevention, to reduce risks and threats. Regarding primary prevention, a continuous work of education about healthy and safe habits could be implemented by the referring physician or health staff. This primary prevention is pivotal from diverse points of view, including health, social and economic consequences for the disabled individual, their close caregivers, and the society in general. Furthermore, as neurorehabilitation has also effects on the long term in different ways according to the individual, it should be part of her/his life project. Secondary prevention is also key since it allows the detection of a disease (comorbidity) in its earliest stages, preventing/limiting a direct impact on autonomy and quality of life of disabled individuals. Working arrangements and professional reorientation should be also included so that disabled workers can return safely to their professional sphere and reintegrate into social life. Lastly, tertiary prevention should aim to soften the impact of the neurological disorder and its lasting effects. This should be done by managing the long-term, often-complex health problems to improve as much as possible their ability to function, their quality of life and their life expectancy in good health. All the above concepts are not new but often scattered and irregularly applied in our societies.

The global approach should take in consideration the patient’s perspectives and wills, reinforcing positive emotional aspects, intrinsic motivation, and reward. Indeed, since motivation enhances motor learning, practice need to be salient and meaningful for optimal engagement and results [133,134]. In this context, the patient is actively involved as her/his training is her/his responsibility. It is important that the therapy’s time and intensity be adapted to the patient’s individual objective/subjective rhythms. Therefore, a set-up of the internal clocks-related performance should be individually regularly checked and adjusted in that way. Crucial aspects of the everyday life should be continuously evaluated, and corrected measures should be taken in case of gap (mood, sleep and rest, diet, mobility, fatigue, motivation, and problem-solving support for ADLs), provided that the noticed gap represents a factual discomfort for the patient. Physical methods should be favored and as far as possible should be preferred to pharmacological surrogate measures. For instance, regular movement and activity can partly attenuate muscle tone disorders (e.g., spasticity), pain, or mood conditions through the modulation of neuroactive endogenous molecules [13,135,136]. Additionally, during physical neurorehabilitation, visuo-motor feedback of the body movements corresponds to the patient’s motor intention and helps the individual to recreate the sensorimotor congruence of the body, restoring its voluntary movement representations and lessening phenomena like the phantom limb syndrome or the hemineglect syndrome [122,137]. The representations of movement and body symmetry are structured in a systemic way through the process of sensorimotor integration [138] and are essential in the neurorehabilitation process, as evidenced by mirror therapy or constraint-induced movement therapy. Since different forms of plasticity occur at different times during training, both eliciting and supporting patients’ functional changes, exercises should be progressive and optimally adapted. In that way, the therapy should be structured to boost skill acquisition and retention. Furthermore, improvement of a function requires an optimal consolidation that is solely obtained through sufficient repetition of the trained action. 

One of the biggest challenges of the neurological disabled person is the accurate control of the action since there is a dissociation between the inputs, the treatment of the information and the resulting outputs. The abnormal body’s spatial representation hinders the exact execution of a task and interferes with vital functions necessary for the individual’s homeostasis (e.g., respiratory function, cardiovascular function, energy metabolism). It is of great importance to support and reinforce the different feedbacks of a disabled-moving body [13,135,136]. This will allow the individual to develop and perfect new strategies for tuning the function and adapt it to the external request. In this sense, the concept of biofeedback can bring to the disabled person during her/his initial neurorehabilitation period a temporary support before the consolidation of new schemes at the same conditions of every single applied technology: meaningfulness for the patient and active participation. It is essential that the provided challenges support learning. Other factors that influence learning and memory, such as emotion and motivation, should be considered as well.

Last but not least, physical and cognitive aids should only be given when really necessary and, more importantly, they should permanently evolve to adapt to the patient’s progression [133]. Indeed, we are not necessarily doing patients a favor by providing them with aids that allow them to accomplish tasks that they cannot quickly accomplish on their own. Of course, the security of the patient is mandatory but, for instance, a wheelchair that impedes the individual in maintaining a certain muscle trophism for sitting postural balance should not be encouraged, at least permanently. Time spent in a wheelchair is time taken away from moving and training (postural balance, verticalization and walking when possible). Additionally, postural defects set in very quickly and when restraint and axial rectification systems are put in place (e.g., seat-brace), very little movement is possible which is devastating for the functional safeguarding of the patient. Similarly, supportive devices can be effective in stabilizing joints and helping patients moving around in everyday life, but they also prevent patients from adequately activating the muscles allowing stabilization of the joints and thus can prevent or delay recovery of function. Regarding active technological aids (robotics, neuroprosthetics), providing passive movement without active patient participation is unlikely to bring major benefits. Robotic aids have evolved and some of them consider the patient’s recovery in allowing her/him to contribute to an increasing extent to the movement as functional recovery progresses. However, if the patient feels the pressure of quickly executing a task and is not motivated or emotionally stable, she/he will leave the robot performing the movement. In this case, the device helping the patient to function better in the short term may end up limiting long-term functional recovery.

Without doubt, a basic concept in neurorehabilitation is learning and resilience. Resilient people have optimistic attitudes and positive emotionality. They have developed coping skills, to effectively balance negative emotions with positive ones. Identifying the factors that influence a person’s resilience is essential in neurorehabilitation. These include the ability to make realistic plans, a positive self-confidence, proper skills for communication and problem-solving, an emotional intelligence, appropriate care and support, and a positive self-image [2,139]. When the individual’s needs are considered in the context of instructions or expectations, the learning/rehabilitation process is significantly accelerated [134].

Practice should be progressive and optimally tailored so that over the course of the program, the demand of the task is adapted in a way that is appropriate to the patient’s abilities and context. It should not be so simple or repetitive as to be unchallenging, nor so difficult as to cause failure in skill acquisition or low self-esteem. Extending skills beyond the context of therapy is crucial in a truly tailored, patient-centered program. This reinforces the gains for “real life” through a virtuous cycle that is more easily assimilated by the patient and self-sustained [2,134].

A final crucial question one should constantly bear in mind is, “Is the person in front of me unable to perform the requested or needed action, or am I the one who does not have the time to wait for them to perform it?” Because after all, whatever the limitation of a neurological patient, movement is the ultimate expression of life no matter how long it takes to achieve it.

## 7. Conclusions

What is finally the risk, for us, of getting back to the human? Over decades, scientists have tried to understand the functioning of the human being throughout the brain. Physicians have always been helpless or even frightened when faced with a patient with a serious neurological deficit. Scientists have constantly developed tools to study these complex entities, functions which are so exceptional but so unknowable, right down to the infinitesimal level. The keys of neural repairing have not been revealed to us by progress, despite many efforts, many clear-sighted and dedicated clinicians, and many brilliant and exceptional researchers. Those keys for neuroregeneration appear to be deadlocked despite the colossal knowledge acquired in ontogenics and multi-omics (genomics, transcriptomics, proteomics, functional lipidomics and metabolomics) and the development of powerful advanced methods of bioinformatics and big data mining [140,141].

The philosophical and human foundations of neurology and neurorehabilitation are impressively summarized by Oliver Sacks [142]: “But it must be said from the outset that a disease is never a mere loss or excess—that there is always a reaction, on the part of the affected organism or individual, to restore, to replace, to compensate for and to preserve its identity, however strange the means may be”.

It is precisely this globality of the being, this silent and ignored effort of an individual to preserve her/his identity and her/his character, this myriad of facets beyond the functional and the dysfunctional, which deserve all our attention. Illness, disease, disability, whether temporary or permanent, are only a few of the various characteristics that shape a person throughout her/his life. However, they are not the only ones, but are part of a whole which constitutes her/his deep nature, her/his human dimension, and which differentiates her/him from others.

The neurorehabilitation process is strewn with pain, restrictions and limits. The other’s gaze and the gaze of oneself are inflexible. Caring in these circumstances requires a constant reevaluation and weighting of the patient’s needs and wants to reestablish the fragile balance that moves her/him, that makes her/him unique beyond the statistics, the natural history of the disease, the incidence and prevalence, and the pathophysiology. The therapeutic program must take into account the patient’s way to reestablish the link with her/his environment, to be proud of what she/he is, of who she/he is, to like the looks that are constantly being cast upon her/him, to manage her/his daily life, however complicated it may be, and regardless of the deviation from the norm. This project is possible and requires a rich and demanding environment; rich, because it involves empathy, goodwill, complicity, discovery, gentleness, harmony or intimacy, and for the patient and therapist to be on equal terms; demanding, because once we leave our comfort zones, we are obliged to trust each other while reconsidering at all times our intention, our gesture, our word, our deepest desire. 

New horizons need to be gradually opened up, and more integrative theories and reflections beyond a certain medical and scientific approach need to emerge. Indeed, a concerted effort to integrate the psychological and social sciences with the physiological sciences will be essential to move forward in concert with recent advances in understanding the importance of patient-centered health care practices and incorporation of close caregivers [4]. We have the sufficient knowledge and expertise in this complex field; we have built the most extraordinary array of tools developed over the years to provide reasonable and reasoned support to people with disabilities. It will therefore be a question of adjusting the use of all these resources to support autonomy for their smooth reintegration in our societies [143,144,145]. Neurorehabilitation is moving forward, and will continue to do so as long as we are willing to take more risks, including that of finally and totally considering the human being, with all that this implies in terms of complexity. We need to build tomorrow’s neurorehabilitation with rigor, method, conviction and exigency, which will allow the rehabilitator to function as a temporary support for those who have been struck down in infancy or adulthood, those for whom the candor of childhood has been destroyed, the dreams of adulthood have been swallowed up, the rich experience of the elderly has been erased. This support will quietly fade away as the individual regains her/his autonomy.

## 8. Study Limitations and Perspectives

The landscape of this narrative review is extremely vast and attempts to cover the scientific literature from the formal beginnings of neurorehabilitation in 1967 [4]. Most studies and reviews on neurorehabilitation focus on their effectiveness and often provide variable results. Moreover, very few encompass the complexity of the neurorehabilitation practice including the appropriate identification of confounding and prognostic factors in order to develop a more personalized evidence-based medicine. In this context, the most significant limit of this study is the gap between the importance of the issues addressed and the possibility to comprehensively summarize the main results in the narrow space of a review. Moreover, since according to the U.S. National Library of Medicine there are more than 600 neurologic diseases, we tried to identify the common denominators of the main diseases in terms of disability, which is somewhat simplistic. Finally, our intention is not to systematically criticize the remarkable developments in the field and our positioning is not a dogma. We believe that a forum for discussion has to be fostered in this area and this is the reason why we intertwined, at our own risk, concrete data relying on scientific references and philosophical considerations based on our day-to-day clinical practice.

## Figures and Tables

**Figure 1 brainsci-12-00982-f001:**
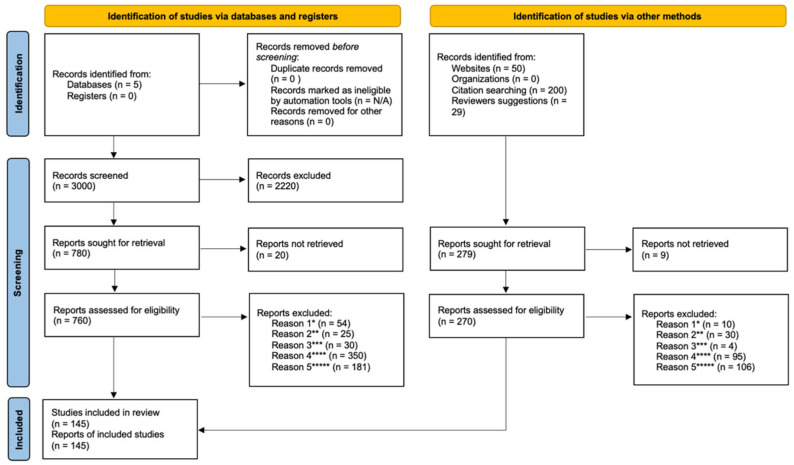
Flow diagram of the systematic review. * Experimental work not confirmed in humans; ** Neurorehabilitation propositions difficult to generalize; *** Theoretical knowledge difficult to extrapolate; **** Works with too small sample size; ***** Works with very low quality evidence of improvement.

## Data Availability

Not applicable.

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
