# Peer review of "After 55 Years of Neurorehabilitation, What Is the Plan?"

_brainsci, 2022, doi:10.3390/brainsci12080982_

Round 1
Reviewer 1 Report
Thank you for providing me the opportunity to review this interesting manuscript. In general, this is a well written and structured manuscript. However, an important issue that is missed by the authors is the psychological factors (especially anxiety and depression as the most common psychiatric conditions in patients with neurological disorders) and their effects on functional outcomes in neurorehabilitation (Please see the following papers for example).
Martens, KA Ehgoetz, et al. "Anxiety is associated with freezing of gait and attentional set-shifting in Parkinson’s disease: a new perspective for early intervention." Gait & posture 49 (2016): 431-436.
Amaricai, Elena, and Dan V. Poenaru. "The post-stroke depression and its impact on functioning in young and adult stroke patients of a rehabilitation unit." Journal of Mental Health 25.2 (2016): 137-141.
Nodehi, Zahra, et al. "Anxiety and cognitive load affect upper limb motor control in Parkinson's disease during medication phases." Annals of the New York Academy of Sciences 1494.1 (2021): 44-58.
Ehgoetz Martens, Kaylena A., Colin G. Ellard, and Quincy J. Almeida. "Anxiety‐provoked gait changes are selectively dopa‐responsive in Parkinson's disease." European Journal of Neuroscience 42.4 (2015): 2028-2035.
Hejazi-Shirmard, Mahnaz, et al. "The effects of anxiety and dual-task on upper limb motor control of chronic stroke survivors." Scientific reports 10.1 (2020): 1-11.
Lee, Eun-Hye, et al. "Association between anxiety and functional outcomes in patients with stroke: a 1-year longitudinal study." Psychiatry Investigation 16.12 (2019): 919.
Paolucci, Stefano, et al. "Post-stroke depression increases disability more than 15% in ischemic stroke survivors: a case-control study." Frontiers in Neurology 10 (2019): 926.
Văcăraș, Vitalie, et al. "The influence of depression and anxiety on neurological disability in multiple sclerosis patients." Behavioural Neurology 2020 (2020).
Ghorbanpour, Zahra, et al. "Overload of anxiety on postural control impairments in chronic stroke survivors: The role of external focus and cognitive task on the automaticity of postural control." Plos one 16.7 (2021): e0252131.
Redmond, Cintasha, et al. "Association of in-hospital depression and anxiety symptoms following stroke with 3 months-depression, anxiety and functional outcome." Journal of Clinical Neuroscience 98 (2022): 133-136.
Haagsma, Juanita A., et al. "Impact of depression and post-traumatic stress disorder on functional outcome and health-related quality of life of patients with mild traumatic brain injury." Journal of neurotrauma 32.11 (2015): 853-862.
Author Response
We thanks the reviewer for their comments that allow to improve the quality and content of our review.
We have included a new chapter (3.4. The impact of psychological factors; page 7 line 340):
3.4. The impact of psychological factors
Psychiatric symptoms, mostly anxiety, depression, and negative thoughts, are concomitant pathologies present in several neurological disorders with variable prevalence. Patients presenting those concomitant pathologies would have an increased risk of mortality, a higher risk of suicidality, increased cognitive impairment, increased risk of falls, increased hospitalization costs, a poorer quality of life, decreased daily functioning and social inter-actions (Paolucci et al., 2019; Ghorbanpour et al., 2021). Also, the results of several studies suggest that those symptoms can directly have a negative impact on the neurological deficits leading to significant deterioration of the functional status of the patient (Paolucci et al., 2019; Prediger et al., 2012; Martens et al., 2014; Ehgoetz Martens et al., 2015; Haagsma et al, 2015; Martens et al., 2016; Văcăraș et al., 2020; Ghorbanpour et al., 2021). However, psychiatric disorders may go unnoticed and have a significant influence on motivation and hence on the rehabilitation process.
For instance, a frequency of post-stroke depression around 30-35% and anxiety around 18-25% has been reported in previous studies (Paolucci et al., 2019; Hackett and Pickles, 2015). The prevalence of anxiety disorders in multiple sclerosis is estimated to be around 30%, three times higher than in the general population. In these patients, the most common forms are generalized anxiety disorder (~18%), panic disorder (~10%), and obsessive-compulsive disorder (~9%) (Văcăraș et al., 2020). Anxiety disorders occur in approximately 25-35% of Parkinson’s patients. Panic disorder, generalized anxiety disorder, and social phobia are the most common anxiety disorders reported in Parkinson’s disease and have been argued to be one of the earliest manifestations of the disease (Martens et al., 2014; Ehgoetz Martens et al., 2015; Martens et al., 2016). Moreover, there is a frequent co-morbidity between anxiety and depressive disorders in those patients, ranging from 14 to 26% (Prediger et al., 2012). Finally, reported prevalence rates of depression following traumatic brain injury varies widely from 17% to 61% depending among others on cause and severity of the brain trauma, and methods used to assess it (Haagsma et al., 2015).
Impaired modulation of neural networks may be responsible for modification of functional outputs through an overload of the capacities of some key brain hubs to process competing inputs coming simultaneously from sensorimotor, cognitive and emotional activity (Martens et al., 2016). Indeed, functional connections between limbic and motor circuits allow emotional inputs to interfere with motor outputs (Ehgoetz Martens et al., 2015). These phenomena could be responsible of gait freezing in Parkinson’s disease (Martens et al., 2014; Ehgoetz Martens et al., 2015; Martens et al., 2016) or postural instability due to improper neuromuscular control in stroke (Ghorbanpour et al., 2021). For instance, anxiety may impair visual information processing as well as the interactions between visual, vestibular, and somatosensory inputs, which are key components of postural control. In addition, high levels of anxiety lead to an attentional bias towards non-task related stimuli, resulting in a decrease in attentional resources allocated to the task and a decrease in performance. (Ghorbanpour et al., 2021).
A key issue in this context is that the impact of psychological factors must be taken at all time into account (and reevaluated periodically) when assessing functional outcomes and implementing different interventions to improve the functional abilities of neurological patients.
Paolucci, S.; Iosa, M.; Coiro, P.; Venturiero, V.; Savo, A.; De Angelis, D.; Morone, G. Post-Stroke Depression Increases Disability More Than 15% in Ischemic Stroke Survivors: A Case-Control Study. Front. Neurol. 2019, 10.
Ghorbanpour, Z.; Taghizadeh, G.; Hosseini, S. A.; Pishyareh, E.; Ghomsheh, F. T.; Bakhshi, E.; Mehdizadeh, H. Overload of Anxiety on Postural Control Impairments in Chronic Stroke Survivors: The Role of External Focus and Cognitive Task on the Automaticity of Postural Control. PloS One 2021, 16 (7), e0252131. https://doi.org/10.1371/journal.pone.0252131.
Prediger, R. D. S.; Matheus, F. C.; Schwarzbold, M. L.; Lima, M. M. S.; Vital, M. A. B. F. Anxiety in Parkinson’s Disease: A Critical Review of Experimental and Clinical Studies. Neuropharmacology 2012, 62 (1), 115–124. https://doi.org/10.1016/j.neuropharm.2011.08.039.
Martens, K. A. E.; Ellard, C. G.; Almeida, Q. J. Does Anxiety Cause Freezing of Gait in Parkinson’s Disease? PLOS ONE 2014, 9 (9), e106561. https://doi.org/10.1371/journal.pone.0106561.
Ehgoetz Martens, K. A.; Ellard, C. G.; Almeida, Q. J. Anxiety-Provoked Gait Changes Are Selectively Dopa-Responsive in Parkinson’s Disease. Eur. J. Neurosci. 2015, 42 (4), 2028–2035. https://doi.org/10.1111/ejn.12928.
Haagsma, J. A.; Scholten, A. C.; Andriessen, T. M. J. C.; Vos, P. E.; Van Beeck, E. F.; Polinder, S. Impact of Depression and Post-Traumatic Stress Disorder on Functional Outcome and Health-Related Quality of Life of Patients with Mild Traumatic Brain Injury. J. Neurotrauma 2015, 32 (11), 853–862. https://doi.org/10.1089/neu.2013.3283.
Martens, K. A. E.; Hall, J. M.; Gilat, M.; Georgiades, M. J.; Walton, C. C.; Lewis, S. J. G. Anxiety Is Associated with Freezing of Gait and Attentional Set-Shifting in Parkinson’s Disease: A New Perspective for Early Intervention. Gait Posture 2016, 49, 431–436. https://doi.org/10.1016/j.gaitpost.2016.07.182.
Văcăraș, V.; Văcăraș, V.; Nistor, C.; Văcăraș, D.; Opre, A. N.; Blaga, P.; Mureșanu, D. F. The Influence of Depression and Anxiety on Neurological Disability in Multiple Sclerosis Patients. Behav. Neurol. 2020, 2020, e6738645. https://doi.org/10.1155/2020/6738645.
Hackett, M. L.; Pickles, K. Part I: Frequency of Depression after Stroke: An Updated Systematic Review and Meta-Analysis of Observational Studies. Int. J. Stroke 2014, 9 (8), 1017–1025. https://doi.org/10.1111/ijs.12357.

Reviewer 2 Report
This is a very interesting Review with a wide perspective aim.
It is well-written and it is interesting to read, however in some parts the argument are not treated in depth and provide personal awesomeness and views of the authors not completely supported by scientific evidences. My comment is a methodological judgment: authors declared to have found more than 3000 papers, read just 200 of them, and only 123 are cited. Some specific sections should deserve a review per se and instead cite few papers, for example: in the section Intensity and effectiveness, a fundamental argument, only 5 papers are cited; in the section The learning Brain only 11 papers are cited, and so on, such ad in Conclusion where it is just a citation of Sacks, claiming some strong concept (like “patient and therapist on equal terms” at line 808) without citing scientific evidences.
I do not like overcitations, and I am not requiring authors to add some references just to increase these numbers, but they should claim the gap between importance of the topics and possibility to summarize the main results in a limited space as the most important limit of the study. Another limit is to have put all together the neurological disorders. Instead a section about the limits of the study, also including that in many parts the personal views of the authors is reported without referring to scientific results, should be reported.
About effectiveness I would suggest authors to take into account the so called "effectiveness paradox" (Morasso et al. 2009; Iosa et al., 2016)
Authors did not take into account the economic burden of neurological rehabilitation, an aspect that often leads the clinical policies (Rajsic et al., 2019; Flachenecker et al. 2019)
At line 401, authors talked about confounding factors, but it is also important to identify prognostic factors (and sometimes there is an overlap among these factors) to develop a more personalized evidence-based medicine
About bio- and neuro-feedback let me suggest you the recent review of Morone et al., 2021
Line 593, Virtual Reality has been not only related to gamification, it was also used in stroke to simulate activities of daily living (Oliveira et al., 2022) or to administer art-therapy (Iosa et al., 2021) or in other tasks not related to gaming.
The paragraph about personalized regenerative medicine (lines 608-644) probably deserves a different title from that related to rehabilitation technologies.
The title is “After 55 years of Neurorehabilitation, what is the plan?”: why 55 years? Authors did not explain the importance of 1967 (I suppose) for the beginning of Neurorehabilitation.
Author Response
We thanks the reviewer for their comments that allow to improve the quality and content of our review.
This is a very interesting Review with a wide perspective aim.
It is well-written and it is interesting to read, however in some parts the argument are not treated in depth and provide personal awesomeness and views of the authors not completely supported by scientific evidences. My comment is a methodological judgment: authors declared to have found more than 3000 papers, read just 200 of them, and only 123 are cited. Some specific sections should deserve a review per se and instead cite few papers, for example: in the section Intensity and effectiveness, a fundamental argument, only 5 papers are cited; in the section The learning Brain only 11 papers are cited, and so on, such ad in Conclusion where it is just a citation of Sacks, claiming some strong concept (like “patient and therapist on equal terms” at line 808) without citing scientific evidences.
Of the 3000 articles identified, we read most of the abstracts (a sentence was added on chapter 2, line 124 : “More than 3000 articles were originally identified between January and June 2022, the abstracts were carefully read, and a shortlist of core published work of potential relevance to this review was prepared.”). After that, we considered mainly the most recent papers about each topic and didn’t include other references that were often included in those papers, as we also do not like over-citations.
I do not like overcitations, and I am not requiring authors to add some references just to increase these numbers, but they should claim the gap between importance of the topics and possibility to summarize the main results in a limited space as the most important limit of the study. Another limit is to have put all together the neurological disorders. Instead a section about the limits of the study, also including that in many parts the personal views of the authors is reported without referring to scientific results, should be reported.
We added the following chapter to the paper according to the reviewer’s comment:
- Study limitations and perspectives
The landscape of this narrative review is extremely vast and attempts to cover the scientific literature from the formal beginnings of neurorehabilitation in 1967 (Harris & Winstein, 2017). Most studies and reviews on neurorehabilitation focus on their effectiveness and often provide variable results. Moreover, very few encompass the complexity of the neurorehabilitation practice including the appropriate identification of confounding and prognostic factors in order to develop a more personalized evidence-based medicine. In this context, the most significant limit of this study is the gap between the importance of the issues addressed and the possibility to comprehensively summarize the main results in the narrow space of a Review. Moreover, since according to the U.S. National Library of Medicine there are more than 600 neurologic diseases, we tried to identify the common denominators of the main diseases in terms of disability which is somehow fairly simplistic. Finally, our intention is not to systematically criticize the remarkable developments in the field and our positioning is not a dogma. We believe that a forum for discussion has to be fostered in this area and this is the reason why we intertwined, at our own risk, concrete data relying on scientific references and philosophical considerations based on our day-to-day clinical practice.
About effectiveness I would suggest authors to take into account the so called "effectiveness paradox" (Morasso et al. 2009; Iosa et al., 2016)
Both references were added in lines 168 and 635 since the concepts discussed in those papers are quite relevant beyond neurorehabilitation through robotics.
Morasso, P.; Casadio, M.; Giannoni, P.; Masia, L.; Sanguineti, V.; Squeri, V.; Vergaro, E. Desirable Features of a “Humanoid” Robot-Therapist. Annu. Int. Conf. IEEE Eng. Med. Biol. Soc. IEEE Eng. Med. Biol. Soc. Annu. Int. Conf. 2009, 2009, 2418–2421. https://doi.org/10.1109/IEMBS.2009.5334954.
Iosa, M.; Morone, G.; Cherubini, A.; Paolucci, S. The Three Laws of Neurorobotics: A Review on What Neurorehabilitation Robots Should Do for Patients and Clinicians. J. Med. Biol. Eng. 2016, 36 (1), 1–11. https://doi.org/10.1007/s40846-016-0115-2.
Authors did not take into account the economic burden of neurological rehabilitation, an aspect that often leads the clinical policies (Rajsic et al., 2019; Flachenecker et al. 2019)
Indeed, this aspect is also paramount in the neurorehabilitation landscape but we consider that it is beyond the scope of this review. We added a sentence concerning this particular point (chapter 4, line 437):
“In the meantime, the magnitude of burden of neurological disorders compels governments and public policy makers to take urgent action to mitigate their risks and impact. However, a number of factors other than clinical efficacy may have an impact on the adoption of innovations and the optimization of care, for example, access to care, cost, medical education and degree of cooperation between different healthcare providers (Feigin et al., 2019; Flachenecker et al. 2019; Rajsic et al., 2019). A clear panorama of this issue is nevertheless beyond the scope of this review.”
At line 401, authors talked about confounding factors, but it is also important to identify prognostic factors (and sometimes there is an overlap among these factors) to develop a more personalized evidence-based medicine.
Prognostic factors were added accordingly in line 21: “In this overly complex panorama, where confounding and prognostic factors also strongly influence potential functional recovery, the healthcare community needs to rethink neurorehabilitation formats.”
About bio- and neuro-feedback let me suggest you the recent review of Morone et al., 2021
The reference of Morone and coworkers was added according to the reviewer’s comment (line 595).
Morone, G.; Ghanbari Ghooshchy, S.; Palomba, A.; Baricich, A.; Santamato, A.; Ciritella, C.; Ciancarelli, I.; Molteni, F.; Gimigliano, F.; Iolascon, G.; Zoccolotti, P.; Paolucci, S.; Iosa, M. Differentiation among Bio- and Augmented- Feedback in Technologically Assisted Rehabilitation. Expert Rev. Med. Devices 2021, 18 (6), 513–522. https://doi.org/10.1080/17434440.2021.1927704.
Line 593, Virtual Reality has been not only related to gamification, it was also used in stroke to simulate activities of daily living (Oliveira et al., 2022) or to administer art-therapy (Iosa et al., 2021) or in other tasks not related to gaming.
The sentence was modified in order to include VR tasks that are not related to gaming (line 649):
“The technology is almost exclusively applied as an adjunct to conventional approaches of cognitive or neuromotor rehabilitation and is typically characterized by the use of gamified exergames which feature game-oriented exercises, although it can also be for example task-oriented in an art-therapy environment (Iosa et al., 2021) or resembling the demands of everyday life activities (Oliveira et al., 2022).”
Iosa, M.; Aydin, M.; Candelise, C.; Coda, N.; Morone, G.; Antonucci, G.; Marinozzi, F.; Bini, F.; Paolucci, S.; Tieri, G. The Michelangelo Effect: Art Improves the Performance in a Virtual Reality Task Developed for Upper Limb Neurorehabilitation. Front. Psychol. 2020, 11, 611956. https://doi.org/10.3389/fpsyg.2020.611956.
Oliveira, J.; Gamito, P.; Lopes, B.; Silva, A. R.; Galhordas, J.; Pereira, E.; Ramos, E.; Silva, A. P.; Jorge, Á.; Fantasia, A. Computerized Cognitive Training Using Virtual Reality on Everyday Life Activities for Patients Recovering from Stroke. Disabil. Rehabil. Assist. Technol. 2022, 17 (3), 298–303. https://doi.org/10.1080/17483107.2020.1749891.
The paragraph about personalized regenerative medicine (lines 608-644) probably deserves a different title from that related to rehabilitation technologies.
Given that the subtitle of the chapter refers to new technologies, regenerative medicine based on stem cell technology seems to fit with that concept. Moreover, even though some of the neurorehabilitation techniques and methodologies presented in this chapter are not new technologies, we chose to put them all under the same umbrella for easy reading.
The title is “After 55 years of Neurorehabilitation, what is the plan?”: why 55 years? Authors did not explain the importance of 1967 (I suppose) for the beginning of Neurorehabilitation.
We added a sentence that explains why 55 years (line 37) “The mid-sixties was the formal starting point for neurorehabilitation.”) as well as the reference about the inaugural STEP conference (and the following: STEP II, III and IV) that originated the neurorehabilitation approach:
Harris, S. R.; Winstein, C. J. The Past, Present, and Future of Neurorehabilitation: From NUSTEP Through IV STEP and Beyond. J. Neurol. Phys. Ther. JNPT 2017, 41 Suppl 3, S3–S9. https://doi.org/10.1097/NPT.0000000000000193.

Reviewer 3 Report
The authors present a narrative review on the status quo and outlooks of neurorehabilitation. The work is well-written and can be a valuable addition to the literature, if the following comments would be addressed.
1) Since the description of the literature search is in good detail, a flow chart may help the readership. https://prisma-statement.org/prismastatement/flowdiagram.aspx
2) The conclusion is too harsh, suggesting current neurorehabilitation to be "away from the human being". (l. 778) The authors should find a phrase that values the success of past approaches and gives room for potential improvement.
3) "The keys of neural repairing have not been revealed to us" (l. 783) - In my understanding, "the keys" are what the authors have laid out quite well in the above sections. Surely, biophysical mechanistic approaches are often blind for patient interaction, however, they are the indispensable foundation of every treatment.
4) The authors cannot wipe away all of their hard work from the beginning of the manuscript by labeling it "mechanistic". (l. 813)
5) Again, the human being has been considered in the literature that the authors describe. (l. 815).
Describe the rationale of "more integrative theories" (l. 812). Please describe holistic approaches as an addition to the existing literature and change the conclusion accordingly. If you think there might be not enough references, this could be a gap analysis for future research.
Author Response
The authors present a narrative review on the status quo and outlooks of neurorehabilitation. The work is well-written and can be a valuable addition to the literature, if the following comments would be addressed.
1) Since the description of the literature search is in good detail, a flow chart may help the readership. https://prisma- statement.org/prismastatement/flowdiagram.aspx
A flowdiagram has been included in the article accordingly (Figure 1; page 3).
2) The conclusion is too harsh, suggesting current neurorehabilitation to be "away from the human being". (l. 778) The authors should find a phrase that values the success of past approaches and gives room for potential improvement.
We have actually valued the amazing successes of past approaches and gave room for potential improvements in the discussion section:
- “Fifty-five years of neurorehabilitation have provided us with amazing tools for helping disabled individuals in their everyday live. However, despite the deep knowledge of the nervous system and the sophisticated devices and methods that we have at our disposal, a ceiling effect is still noticed with respect to functional recovery.” (line 723)
- “Bearing in mind the different aspects likely to interfere in the rehabilitation process and the complexity of the many overlapping aspects of human being, the neurorehabilitation of the future is probably far more human than expected, even if it relies on the stronger points of neurorehabilitation methods and innovative developments.” (line 735)
- “This will allow the individual to develop and perfect new strategies for tuning the function and adapt it to the external request. In this sense, the concept of biofeedback can bring to the disabled person during her/his initial neurorehabilitation period a temporarily support before the consolidation of new schemes at the same conditions of every single applied technology: meaningfulness for the patient and active participation.” (line 791)
- “Robotic aids have evolved and some of them consider the patient’s recovery allowing her/him to contribute to an increasing extent to the movement as functional recovery progresses.” (line 814)
The sentence “Is the person in front of me unable to perform the requested or needed action, or am I the one who doesn't have the time to wait for them to perform it?” (line 837) is a general consideration which challenges the relationship between the disabled person and the valid one (not necessarily the therapist or the physician).
Our intention is not to systematically criticize the remarkable developments in the field and our positioning is not a dogma. We believe that a forum for discussion has to be fostered in this area and this is the reason why we intertwine, at our own risk, concrete data relying on scientific references and philosophical considerations based on our day-to-day clinical practice. The conclusion is indeed the result of our thoughts and feedbacks we regularly receive about the practice and generally speaking about care and quality of life of patients.
However, we have added a sentence in the Conclusion that values the positive points of past approaches (line 885) according to the reviewer’s comment:
“We have the sufficient knowledge and expertise in this complex field; we have built the most extraordinary array of tools developed over the years to provide reasonable and reasoned support to people with disabilities. It will therefore be a question of adjusting the use of all these resources to support autonomy for their smooth reintegration in our societies.”
3) "The keys of neural repairing have not been revealed to us" (l. 783) - In my understanding, "the keys" are what the authors have laid out quite well in the above sections. Surely, biophysical mechanistic approaches are often blind for patient interaction, however, they are the indispensable foundation of every treatment.
We referred here to neuroregeneration pathways since the current state-of-the-art appears to be deadlocked despite the colossal knowledge acquired in ontogenics and multi-omics (genomics, transcriptomics, proteomics, functional lipidomics, metabolomics) and the development of powerful advanced methods of bioinformatics and big data mining.
We clarified our purpose (line 850) and gave some references:
“Those keys for neuroregeneration appear to be deadlocked despite the colossal knowledge acquired in ontogenics and multi-omics (genomics, transcriptomics, proteomics, functional lipidomics, metabolomics) and the development of powerful advanced methods of bioinformatics and big data mining (Baron et al., 2015; Yasuhara et al., 2020)”
Baron, R.; Ferriero, D. M.; Frisoni, G. B.; Bettegowda, C.; Gokaslan, Z. L.; Kessler, J. A.; Vezzani, A.; Waxman, S. G.; Jarius, S.; Wildemann, B.; Weller, M. Neurology--the next 10 Years. Nat. Rev. Neurol. 2015, 11 (11), 658–664. https://doi.org/10.1038/nrneurol.2015.196.
Yasuhara, T.; Kawauchi, S.; Kin, K.; Morimoto, J.; Kameda, M.; Sasaki, T.; Bonsack, B.; Kingsbury, C.; Tajiri, N.; Borlongan, C. V.; Date, I. Cell Therapy for Central Nervous System Disorders: Current Obstacles to Progress. CNS Neurosci. Ther. 2020, 26 (6), 595–602. https://doi.org/10.1111/cns.13247.
4) The authors cannot wipe away all of their hard work from the beginning of the manuscript by labeling it "mechanistic". (l. 813)
We have removed the word "mechanistic" to soften our statement.
5) Again, the human being has been considered in the literature that the authors describe. (l. 815).
Our intention is not to systematically criticize the remarkable developments in the field and our positioning is not a dogma. We believe that a forum for discussion has to be fostered in this area and this is the reason why we intertwine, at our own risk, concrete data relying on scientific references and philosophical considerations based on our day-to-day clinical practice. The conclusion is indeed the result of our thoughts and feedbacks we regularly receive about the practice and generally speaking about care and quality of life of patients.
Describe the rationale of "more integrative theories" (l. 812). Please describe holistic approaches as an addition to the existing literature and change the conclusion accordingly. If you think there might be not enough references, this could be a gap analysis for future research.
We added the following sentence with respect to the rationale (line 881):
“Indeed, a concerted effort to integrate the psychological and social sciences with the physiological sciences will be essential to move forward in concert with recent advances in understanding the importance of patient-centered health care practices and incorporation of close caregivers (Harris & Winstein, 2017).”
Harris, S. R.; Winstein, C. J. The Past, Present, and Future of Neurorehabilitation: From NUSTEP Through IV STEP and Beyond. J. Neurol. Phys. Ther. JNPT 2017, 41 Suppl 3, S3–S9. https://doi.org/10.1097/NPT.0000000000000193.
We also added three new references concerning the concept of holistic neurorehabilitation (line 890):
Kapur, N. Paradoxes in Rehabilitation. Disabil. Rehabil. 2020, 42 (11), 1495–1502. https://doi.org/10.1080/09638288.2019.1572795.
Mane, R.; Chouhan, T.; Guan, C. BCI for Stroke Rehabilitation: Motor and Beyond. J. Neural Eng. 2020, 17 (4), 041001. https://doi.org/10.1088/1741-2552/aba162.
Bradnam, L. V.; Meiring, R. M.; Boyce, M.; McCambridge, A. Neurorehabilitation in Dystonia: A Holistic Perspective. J. Neural Transm. Vienna Austria 1996 2021, 128 (4), 549–558. https://doi.org/10.1007/s00702-020-02265-0.

Round 2
Reviewer 1 Report
In their replies, the authors have satisfactorily addressed my comment.
Reviewer 2 Report
I appreciated your effort in reviewing the manuscript for taking into account my suggestions and I am satisfied about the performed changes
Reviewer 3 Report
The authors made thoughtful changes throughout the whole manuscript, so that it will be a valuable addition to the literature.